# New Insights of OLFM2 and OLFM4 in Gut-Liver Axis and Their Potential Involvement in Nonalcoholic Fatty Liver Disease

**DOI:** 10.3390/ijms23137442

**Published:** 2022-07-04

**Authors:** Laia Bertran, Rosa Jorba-Martin, Andrea Barrientos-Riosalido, Marta Portillo-Carrasquer, Carmen Aguilar, David Riesco, Salomé Martínez, Margarita Vives, Fàtima Sabench, Daniel Del Castillo, Cristóbal Richart, Teresa Auguet

**Affiliations:** 1Grup de Recerca GEMMAIR (AGAUR)-Medicina Aplicada, Departament de Medicina i Cirurgia, Universitat Rovira i Virgili (URV), Institut d’Investigació Sanitària Pere Virgili (IISPV), 43007 Tarragona, Spain; laia.bertran@urv.cat (L.B.); andreitabarri18@gmail.com (A.B.-R.); marta.portillo.carrasquer@gmail.com (M.P.-C.); caguilar.hj23.ics@gencat.cat (C.A.); cristobalmanuel.richart@urv.cat (C.R.); 2Servei de Cirurgia General i de l’Aparell Digestiu, Hospital Universitari de Tarragona Joan XXIII, Universitat Rovira i Virgili (URV), Institut d’Investigació Sanitària Pere Virgili (IISPV), Mallafré Guasch, 4, 43007 Tarragona, Spain; rjorba.hj23.ics@gencat.cat; 3Servei Medicina Interna, Hospital Universitari de Tarragona Joan XXIII, Mallafré Guasch, 4, 43007 Tarragona, Spain; david_riesco@hotmail.com; 4Servei Anatomia Patològica, Hospital Universitari de Tarragona Joan XXIII, Mallafré Guasch, 4, 43007 Tarragona, Spain; mgonzalez.hj23.ics@gencat.cat; 5Servei de Cirurgia, Hospital Sant Joan de Reus, Departament de Medicina i Cirurgia, Universitat Rovira i Virgili (URV), Institut d’Investigació Sanitària Pere Virgili (IISPV), Avinguda Doctor Josep Laporte, 2, 43204 Reus, Spain; mvives@gmail.com (M.V.); fatima.sabench@urv.cat (F.S.); danieldel.castillo@urv.cat (D.D.C.)

**Keywords:** nonalcoholic fatty liver disease, gut-liver axis, obesity, olfactomedin

## Abstract

Olfactomedins (OLFMs) are a family of glycoproteins that play a relevant role in embryonic development and in some pathological processes. Although OLFM2 is involved in the regulation of the energy metabolism and OLFM4 is an important player in inflammation, innate immunity and cancer, the role of OLFMs in NAFLD-related intestinal dysbiosis remains unknown. In this study, we analysed the hepatic mRNA expression of *OLFM2* and the jejunal expression of *OLFM4* in a well-established cohort of women with morbid obesity (MO), classified according to their hepatic histology into normal liver (*n* = 27), simple steatosis (*n* = 26) and nonalcoholic steatohepatitis (NASH, *n* = 16). Our results showed that *OLFM2* hepatic mRNA was higher in NASH, in advanced degrees of steatosis and in the presence of lobular inflammation. Additionally, we obtained positive correlations between hepatic *OLFM2* and glucose, cholesterol, trimethylamine N-oxide and deoxycholic acid levels and hepatic fatty acid synthase, and negative associations with weight and jejunal Toll-like receptors (*TLR4*) and *TLR5* expression. Regarding jejunal *OLFM4*, we observed positive correlations with circulating interleukin (IL)-8, IL-10, IL-17 and jejunal *TLR9*. In conclusion, OLFM2 in the liver seems to play a relevant role in NAFLD progression, while OLFM4 in the jejunum could be involved in gut dysbiosis-related inflammatory events.

## 1. Introduction

Nonalcoholic fatty liver disease (NAFLD) is characterized by ectopic fat accumulation in hepatic tissue in the absence of secondary causes of steatosis, such as alcohol intake and/or viral infection [1,2]. NAFLD is a general term used for a variety of liver conditions that comprises many stages from simple steatosis (SS) to nonalcoholic steatohepatitis (NASH), this last one defined by the presence of hepatic inflammation, ballooning and/or fibrosis [3,4], which can trigger cirrhosis or hepatocellular carcinoma if left untreated [5]. Despite NAFLD being the most extended chronic hepatic disorder and the existence of many studies characterizing it, its underlying molecular mechanisms remain incompletely understood, and a specific treatment is not currently approved. Thus, understanding the pathogenic processes behind NAFLD provides the basis for identifying the onset and progression of the disease [6].

The severity of NAFLD is related to different comorbidities, such as obesity and type 2 diabetes mellitus (T2DM), and recently it was also related to intestinal dysbiosis [7]. The intestinal microbiota seem to play an important role in human health, since it is known to be altered in pathological conditions, leading to intestinal dysbiosis [8,9]. The altered microbiota trigger an increase in the permeability of the intestinal barrier, which induces the release of microbiota-derived mediators such as lipopolysaccharides (LPS), short-chain fatty acids (SCFAs), choline metabolites, bile acids (BAs) and endogenous ethanol [10]. Recently, the relevance of the gut-liver axis and its communication through the portal vein has been described. When the permeability of the intestinal barrier is corrupted, the liver is exposed to components of the microbiota contributing to inflammation and liver injury and thus to the pathogenesis of NAFLD [11].

In this regard, the gut-liver axis [12] and other crosstalk systems, such as the trade-off of adipokines in the adipose tissue-liver axis [13], have reinforced the fact that NAFLD is a multisystemic condition influenced by metabolic and inflammatory processes occurring in related tissues [14]. Thus, evaluating NAFLD as a disorder with hepatic and extrahepatic manifestations [15] would help us to understand the physiological processes involved in this disease.

The seven groups (I–VII) of the olfactomedin (OLFM) family of glycoproteins are located in C-terminal regions, except those from groups II and VI. OLFMs play a key role in cell–cell signalling and cell adhesion and differentiation, which suggests involvement in many molecular mechanisms of cancer [16]. Although the biological functions of OLFM domain-containing proteins remain mostly elusive, continuous evidence indicates that these proteins may play relevant roles in normal development and pathological processes [17]. Additionally, the participation of OLFMs in haematopoiesis and the early development of the nervous system has been reported. Additionally, OLFMs have been associated with diseases, such as gastrointestinal cancer and glaucoma, since the interruption of their expression leads to the development of disturbances and lethality [16].

On the one hand, olfactomedin 2 (OLFM2; also known as noelin-2) is a secretory glycoprotein that is a member of OLFM group I together with OLFM1 and 3. This protein is mainly expressed in neurons, but also in the liver [18] and adipose tissue [18,19]. Moreover, González-García et al. demonstrated that OLFM2 is involved in the regulation of energy metabolism [20].

On the other hand, olfactomedin 4 (OLFM4) belongs to OLFM group V. This glycoprotein was found to be endogenously expressed in bone marrow and the small intestine, stomach, colon, prostate [21] and pancreas [22]. In addition, this protein has been related to multiple signalling pathways and factors, including the NF-κB [23] WNT/β-catenin pathway [24,25] and Notch proteins [26,27], which are strongly related to pathological processes in hepatic diseases, such as NAFLD [28,29,30]. Furthermore, OLFM4 is an important player in the cellular process of inflammation, and it has important roles in innate immunity against bacterial infection, gastrointestinal inflammation, and cancer [17,31].

OLFM2 seems to have a role in the regulation of the lipid metabolism, and OLFM4 may participate in gastrointestinal inflammation in relation to dysbiosis. Furthermore, the Human Protein Atlas has reported OLMF2 expression in the liver and OLFM4 expression in the small intestine [18,19]. Thus, the current study aimed to analyse *OLFM2* mRNA expression in liver biopsies and *OLFM4* mRNA expression in jejunum samples in a well-established cohort of women with morbid obesity (MO) with different NAFLD grades to explore the role of OLFMs in the pathogenesis of NAFLD.

## 2. Results

### 2.1. Baseline Characteristics of Subjects

The clinical and biochemical measurements of the cohort formed by women with MO (BMI > 40 kg/m^2^) classified according to the hepatic histology as normal liver (NL, *n* = 27), SS (*n* = 26) and NASH (*n* = 16), are shown in Table 1. The participants did not present significant differences between groups in terms of weight, body mass index (BMI), systolic blood pressure (SBP), diastolic blood pressure (DBP), homeostatic model assessment method-insulin resistance (HOMA1-IR), insulin, glycosylated haemoglobin (HbA1c), cholesterol, high density lipoprotein cholesterol (HDL-C), low density lipoprotein cholesterol (LDL-C), aspartate-aminotransferase (AST), alanine-aminotransferase (ALT) and gamma-glutamyltransferase (GGT). However, in this analysis, we found higher levels of glucose and alkaline phosphatase (ALP) in the SS group than in the NL group, increased levels of triglycerides in the NASH cohort compared to NL women and increased levels of ALP in SS subjects in comparison to NASH patients.

### 2.2. Evaluation of the Relative mRNA Abundance of OLFM2 and OLFM4 According to Hepatic Histology

To achieve the aim of this study, that was to explore the role of OLFMs in NAFLD progression, we evaluated the *OLFM2* mRNA expression in liver samples in a cohort of women with MO. Moreover, given that the importance of the gut-liver axis in NAFLD progression has been reported, we also analysed the jejunal *OLFM4* mRNA expression.

First, when we analysed *OLFM2* and *OLFM4* relative mRNA expressions between NL and NAFLD in hepatic and jejunal samples, respectively, we only reported significant differences between groups in the case of *OLFM2* (*p* = 0.005). Then, we subclassified the patients according to the hepatic histopathological grades into NL, SS and NASH. In this sense we found a higher expression of hepatic *OLFM2* in SS and NASH women compared to the NL cohort (Figure 1A). Later, when the subjects were divided according to the presence of NASH, an increase in *OLFM2* relative expression in the NASH group was observed in comparison with the non-NASH group, as shown in Figure 1B. However, regarding *OLFM4*, we did not report significant differences between groups (Figure 1C,D).

### 2.3. Evaluation of the Relative mRNA Abundance of OLFM2 and OLFM4 According to the Severity of Steatosis

To deepen the knowledge of the link between OLFMs and NAFLD, first we wanted to focus on the hepatic lipid content degree, so we classified the cohort into different grades of steatosis. In this sense, we reported an enhanced expression of hepatic *OLFM2* mRNA in moderate and severe stages of steatosis in comparison to those subjects without steatosis (Figure 2A); while we did not find significant differences when *OLFM4* was analysed, as graphically represented in Figure 2B.

### 2.4. Evaluation of the Relative mRNA Abundance of OLFM2 and OLFM4 According to NASH-Related Parameters

Later, we wanted to focus the analysis on the main parameters of the advanced stage of NAFLD, such as inflammation and hepatocellular ballooning; we therefore evaluated their link with OLFM2 and OLFM4. Concerning portal inflammation, we did not observe differential expressions of *OLFM2* in the liver and *OLFM4* in the jejunum between groups (Figure 3A,B). Regarding lobular inflammation, we found an increase in *OLFM2* hepatic expression in the cohort presenting this type of inflammation (Figure 3C), but we did not find differences with regard to *OLFM4* jejunal expression (Figure 3D). Moreover, non-significant differences in *OLFM2* and *OLFM4* were reported when the cohort was classified based on the presence of hepatocyte ballooning (Figure 3E,F).

### 2.5. Correlations of Relative mRNA Abundance of Hepatic OLFM2 and Jejunal OLFM4, with Clinical and Biochemical-Related Parameters

To broaden the study of OLFM2, we analysed their correlations with different parameters related to glucose and lipid metabolism, gut microbiota and inflammation in a cohort of women with MO. First, we observed positive correlations between hepatic *OLFM2* mRNA expression and glucose, cholesterol, trimethylamine N-oxide (TMAO), deoxycholic acid (DCA) levels and fatty acid synthase (*FAS*) mRNA expression in the liver, as shown in Figure 4A–E, respectively. We also found a negative association between hepatic *OLFM2* mRNA expression and weight, jejunal *TLR4* and jejunal *TLR5* expression, as was graphically represented in Figure 4F–H.

On the other hand, when we focused on the *OLFM4* in the jejunum, we observed positive correlations between *OLFM4* and circulating interleukin (IL)-8, IL-10, IL-17 levels and jejunal *TLR9* expression, as shown in Figure 5A–D, respectively.

## 3. Discussion

This study is important as it is the first to report an interesting association between OLFMs and NAFLD in hepatic and jejunal samples of women with MO. It is already known that more in-depth studies of the molecular mechanisms involved in NAFLD are necessary to find new therapeutic targets. In this sense, given that OLFMs are strongly linked to cellular processes [16,17] and have been related to some disorders, such as obesity and hepatocarcinoma (HCC), among others [20,31,32,33,34], we wanted to evaluate the possible roles of OLFM2 and OLFM4 in NAFLD pathogenesis.

Regarding our first finding, when patients were classified according to their hepatic histopathology, *OLFM2* mRNA expression in the liver increased when hepatic health worsened. Moreover, a significant increase in *OLFM2* expression in the presence of NASH was observed when the cohort was classified into non-NASH and NASH groups. These facts seem to suggest that OLFM2 is involved in NAFLD progression, especially in NASH. Although *OLFM2* mRNA abundance has been found in liver tissues [18], its expression has not been previously evaluated, and this protein has not been related to NAFLD. Therefore, OLFM2 in liver tissues seems to play a role in NAFLD progression in subjects with obesity, but further studies are needed in this field.

Then, when we evaluated hepatic *OLFM2* and jejunal *OLFM4* based on the steatosis stage of the disease, we found a significant increase in hepatic *OLFM2* in patients with moderate and severe steatosis grades. Later, when we analysed the role of these OLFMs in accordance with steatohepatitis parameters, such as lobular and portal inflammation and hepatocyte ballooning, we found an increase in hepatic *OLFM2* expression in patients with lobular inflammation. These findings indicate that *OLFM2* increases as the disease progresses, since it is greatly increased in NASH, which is usually characterized by hepatic inflammation and elevated lipid content [3,35]. As we have previously mentioned, this is a novel finding; thus, no previous reports exist in this regard, not even reports on the expression of this protein in the liver. More studies are necessary to corroborate this evidence.

Unfortunately, although OLFM4 seems to play relevant roles as an anti-inflammatory molecule in *Helicobacter pylori* infection [31] and gastric and intestinal disorders [33,34], which could suggest that OLFM4 is involved in NAFLD pathogenesis through microbiota changes and the gut-liver axis, in our study, the mRNA expression of this protein in jejunal samples was not significantly different between the hepatic histopathological groups. In the literature there are no other studies in this area, but probably in those patients with MO who present an intestinal dysbiosis associated with obesity, there is an activation of inflammasomes and also of the immune system [36,37] that gives rise to an inflammatory microenvironment capable of masking the possible relationship that could have expression in the jejunum of *OLFM4* with liver histology. Therefore, to further investigate this relationship, additional studies should be performed in subjects of normal weight.

To deepen the knowledge of OLFMs in regard to NAFLD, we analysed several correlations. Positive associations between hepatic *OLFM2* mRNA and hepatic *FAS* mRNA expression and cholesterol and glucose levels, which are parameters closely related to NAFLD, were found [38,39]. When there is an excess of carbohydrates in the diet, glucose can be stored as lipids in the liver, and FAS, a lipogenic enzyme, synthesizes triglycerides from free fatty acids; therefore, its action enhances hepatic steatosis, the principal characteristic of NAFLD [39,40]. In addition, another study reported increased *FAS* mRNA expression and high glucose levels in people with NAFLD [41], so our current results are consistent with previously mentioned publications showing dysregulation of hepatic lipogenesis in NAFLD [42,43]. All this information, as well as our findings of positive correlations of *OLFM2* in the liver with glucose and cholesterol levels and hepatic *FAS*, corroborates our previous hypothesis that hepatic expression of *OLFM2* increases as NAFLD progresses, when at the same time, the metabolic imbalance also worsens [44].

Additionally, we found a negative significant correlation between hepatic OLFM2 and weight. This association is difficult to explain, given that González-García et al. observed that a global lack of OLFM2 induces weight loss [20]. This contradiction could be explained by the fact that our cohort presented a degree of severe obesity (BMI between 40 and 46 Kg/m^2^), and perhaps some correlations may have been skewed due to such excessive BMI.

On the other hand, we also observed a positive correlation of hepatic *OLFM2* with DCA and TMAO levels. TMAO is a gut microbiota-dependent metabolite, and some publications [42,45,46] reported increased TMAO levels in patients with NAFLD. The same authors suggested that TMAO could contribute to the development and severity of this disease due to its role in glucose regulation, cholesterol homeostasis and lipid absorption [45,47]. An increase in DCA levels, which are involved in the metabolism of ingested lipids, was also seen in NAFLD patients [48]. Additionally, Grzych et al. showed high levels of DCA in the presence of NASH [49]. Therefore, these positive associations of hepatic *OLFM2* with DCA and TMAO levels remain consistent with previous results, reinforcing the hypothesis that OLFM2 may play an important role in the progression of NAFLD.

Later, we wanted to analyse whether OLFM2 in the liver is associated with jejunal TLRs, since these are involved in gut-liver axis crosstalk that can influence NAFLD appearance and progression [50]. In this regard, although we previously observed that hepatic *OLFM2* mRNA expression is enhanced when the disease becomes severe, we observed a negative correlation between jejunal *TLR4* and *5* with hepatic *OLFM2* mRNA expression. This is a contradictory result given that TLRs are known to be involved in NAFLD progression [50], but this result may be because women in our cohort followed a very low-calorie diet for 3 weeks before bariatric surgery [51]. In this sense, Macedo Rogero et al. demonstrated that omega-3 polyunsaturated fatty acids, usually included in the very low-calorie diet [52], attenuate the activation of the TLR4 signalling pathway, exerting an anti-inflammatory effect [53] and inhibiting lipid accumulation [54]. Concerning TLR5, this receptor has been shown to not necessarily be altered by diet [55]. Otherwise, TLR5 seems to play a key role in liver protection against intestinal dysbiosis-induced NAFLD in mice [56], preventing gut inflammation-related disorders [57]. Moreover, an increase in TLR5 seems to contribute to liver regeneration events [58]. Hence, in this case, the negative correlation of *TLR5* with *OLFM2* mRNA expression in the liver makes sense given that *OLFM2* increases as NAFLD worsens, while *TLR5* in the gut could be decreased due to the high inflammation pattern induced by intestinal dysbiosis and liver damage. Similarly, mouse models without TLR5 expression were used to present obesity, hepatic steatosis and insulin resistance, the main characteristics of our study cohort [59].

Focusing on *OLFM4*, as stated above, its relative mRNA expression in the jejunum did not present significant differences between NAFLD groups, but we did observe positive associations with IL-8, IL-10 and IL-17 levels and jejunal *TLR9* expression. In this sense, circulating IL-8 and IL-17 are defined as proinflammatory cytokines that present enhanced circulating levels in NAFLD [60,61,62]. Therefore, in this work, OLFM4 seems to be related to a proinflammatory state, despite previously being linked to an anti-inflammatory role in other tissues [31,63]. Hence, we cannot definitively conclude that OLFM4 in the jejunum is involved in gut dysbiosis-related inflammation because our patients presented low-grade chronic inflammation due to obesity [64], but it would be interesting to evaluate *OLFM4* jejunal abundance in lean subjects.

IL-10 is a well-defined anti-inflammatory cytokine [65] with protective effects in liver injury [66], but high levels of circulating IL-10 have sometimes been reported in subjects with obesity [67]. Given that IL-10 attenuates the secretion of proinflammatory cytokines, a continuous increase in IL-10 levels could be explained by the competition of IL-10 with proinflammatory cytokines in an attempt to balance the inflammatory state of these patients [67]. Moreover, IL-10 exhibits feedback regulation to inhibit proinflammatory cytokine production [68]. These findings could explain the positive correlation between *OLFM4* mRNA in the jejunum, which was previously correlated with proinflammatory cytokines, and IL-10 circulating levels, since this anti-inflammatory cytokine seems to counteract the low-grade chronic inflammatory pattern of our MO patients.

TLR9 is a receptor that is closely linked to metabolism and inflammatory events [69], and it has been suggested it plays a relevant role in NAFLD pathogenesis [50]. Additionally, it was clinically demonstrated that TLR9 is a key driver of NASH [69,70]. Hence, the positive association of *TLR9* with *OLFM4* jejunal expression also makes sense, as it is consistent with its correlation with proinflammatory cytokines. This finding reinforces the possible inflammatory role of OLFM4 in the jejunum, but more studies are needed.

In summary, we have carried out a novel study showing that the hepatic expression of *OLFM2* seems to increase as NAFLD progresses, and this protein seems to be associated with NASH and with a severe steatosis pattern in NAFLD patients. Additionally, *OLFM4* was suggested to be more related to inflammation that occurs due to intestinal dysbiosis, usually linked to NAFLD. In this sense, we are pioneers in demonstrating a relationship between OLFM2 and NAFLD and between OLFM4 and inflammation related to gut dysbiosis. However, there are some limitations to this study; for example, in the literature, there is little or no experience in this field. In addition, this is a cross-sectional study, so a causal relationship cannot be confirmed. Lastly, a cohort of patients made up of only MO women was used, so these results cannot be extrapolated to other sexes or other groups of people with obesity, overweight or normal weight. Hence, future studies with a larger cohort and assessing the role of these proteins in lean subjects with NAFLD, are needed to understand the specific role of these OLFMs in the pathogenesis of NAFLD and their relationship with the gut-liver axis. Moreover, to analyse OLFMs expression in in vivo and in vitro models of liver lipid overload would also be necessary. In addition, it could also be interesting to evaluate the *OLFM4* expression in other sections of the gastrointestinal tract to elucidate its involvement in gut inflammation and its relationship to NAFLD.

## 4. Materials and Methods

### 4.1. Subjects

The institutional review board (Institut Investigació Sanitària Pere Virgili (IISPV) CEIm; 23c/2015; 11 May 2015) approved this research. Informed written consent was obtained from all participants. The cohort was formed of 69 Caucasian women with MO (BMI > 40 kg/m^2^). Liver and jejunum biopsies were collected during a planned laparoscopic bariatric surgery and liver samples were indicated for clinical diagnosis. The exclusion criteria were as follows: (1) an intake of ethanol higher than 10 g/day or other toxins; (2) patients who had infectious disease or neoplastic disease or an acute or chronic hepatic disease different from NAFLD; (3) menopausal women or women using contraceptives in order to avoid the interference of hormones which can cause biases in glucose and lipid metabolism as well as in cytokine determinations; (4) patients treated with fibrates because this treatment can interfere with the metabolism of some microbiota-derived metabolites studied in this work.

### 4.2. Sample Size

The work was mainly focused on defining the specific role of OLFM2 in hepatic tissue and OLFM4 in jejunum samples in MO patients with or without NAFLD. To achieve our objective, sample sizes were calculated using a GRANMO calculator accepting an alpha risk of 0.05 and a beta risk of less than 0.2 in a bilateral contrast; 25 subjects in the first group (NL: control group) and 50 in the second (NAFLD: SS and NASH) are needed to detect as statistically significant the difference between two proportions, which for group 1 is expected to be of 0.33 and group 2 of 0.67. The ARCSINUS approach was used.

### 4.3. Liver Pathology

Liver samples were classified by the method described elsewhere [71,72], using haematoxylin and eosin and Masson’s trichrome stains. They were then scored by an experienced hepatopathologist. According to their hepatic histopathology, women with MO were classified into NL (*n* = 27) and NAFLD (*n* = 42). Patients with NAFLD were subclassified SS (micro/macrovesicular steatosis without inflammation or fibrosis, *n* = 26), and NASH (Brunt grades 1–2, *n* = 16). None of the patients with NASH in our cohort presented fibrosis.

### 4.4. Biochemical Analyses

Physical, anthropometric, and biochemical evaluation were performed on all the studied cohort. Blood samples were extracted through a BD Vacutainer^®^ system by specialized nurses, after overnight fasting and before bariatric surgery. Venous blood samples were obtained in tubes with or without ethylenediaminetetraacetic acid, which were separated in plasma and serum aliquots by centrifugation (3500 rpm, 4 °C, 15 min). Conventional automated analyser was used to analyse biochemical parameters. IR was estimated using homeostatic model assessment for IR (HOMA1-IR). Cytokines, such as interleukin (IL)-1β, IL-6, IL-8, IL-10, IL-17, TNF-α and adiponectin, were determined using multiplex sandwich immunoassays and the MILLIPLEX MAP Human Adipokine Magnetic Bead Panel 1 (HADK1MAG-61K, Millipore, Billerica, MA, USA), the MILLIPLEX MAP Human High-Sensitivity T Cell Panel (HSTCMAG28SK, Millipore, Billerica, MA, USA), and the Bio-Plex 200 instrument, according to the manufacturer’s instructions. Absolute quantification of circulating bile acids, choline, trimethylamine (TMA), trimethylamine N-oxide (TMAO), betaine and short-chain fatty acids were analysed by liquid chromatography coupled to triple-quadrupole-mass spectrometry (LC-QqQ). All these analyses were assessed at the Center for Omic Sciences (Rovira i Virgili University-Eurecat).

### 4.5. mRNA Expression in Liver and Jejunum

Hepatic and jejunal samples were collected during bariatric surgery and conserved in tubes with RNAlater (Qiagen, Hilden, Germany) at 4 °C. Samples were then processed and stored at −80 °C. RNeasy mini kit (Qiagen, Barcelona, Spain) was used to extract total RNA from the liver and the jejunum. Reverse transcription to cDNA was performed with the High-Capacity RNA-to-cDNA Kit (Applied Biosystems, Madrid, Spain). Real-time quantitative PCR was carried out with the TaqMan Assay predesigned by Applied Biosystems for the detection of *OLMF2* (Hs01017934_m1) mRNA in the liver and *OLFM4* (Hs00197437_m1) mRNA in the jejunum; we also evaluated the mRNA of some hepatic lipid metabolism-related genes such as sterol-regulatory-element-binding protein 1c (*SREBP1c*) (Hs01088691_m1), liver X receptor alpha (*LXRα*) (Hs00173195_m1), fatty acid synthase (*FAS*) (Hs00188012_m1); and TLRs in jejunum (*TLR2* (Hs02621280_s1), *TLR4* (Hs00152939_m1), *TLR5* (Hs05021301_s1), *TLR9* (Hs00370913_s1)). The expression of each gene was calculated standardized to the mRNA expression of *18S RNA* (Fn04646250_s1) for hepatic genes, and glyceraldehyde-3-phosphate dehydrogenase (*GAPDH*) (Hs02786624_g1) for jejunal genes, after they were normalized using the control group (NL) as a reference. All reactions were duplicated in 96-well plates using the QuantStudio™ 7 Pro Real-Time PCR System (Applied Biosystem, Foster City, CA, USA).

### 4.6. Statistical Analysis

The data were analysed using the SPSS/PC+ for Windows statistical package (version 27.0; SPSS, Chicago, IL, USA). The distribution of variables was obtained using the Kolmogorov–Smirnov test. All results were expressed as the median and the interquartile range (25th–75th). The different comparative analyses were assessed using Mann–Whitney U test to compare groups. The coefficient of correlation (rho) between variables was calculated using Spearman’s method. *p*-values < 0.05 were statistically significant. Graphics were elaborated using GraphPad Prism software (version 7.0; GraphPad, San Diego, CA, USA).

## 5. Conclusions

In conclusion, the first novel finding of this study is that *OLFM2* expression in the liver seems to play a relevant role in NAFLD progression in women presenting MO. Secondly, our results suggested that jejunal *OLFM4* expression could be involved in gut dysbiosis-related inflammatory events in women with MO. In any case, more research in this field is necessary to confirm these hypotheses.

## Figures and Tables

**Figure 1 ijms-23-07442-f001:**
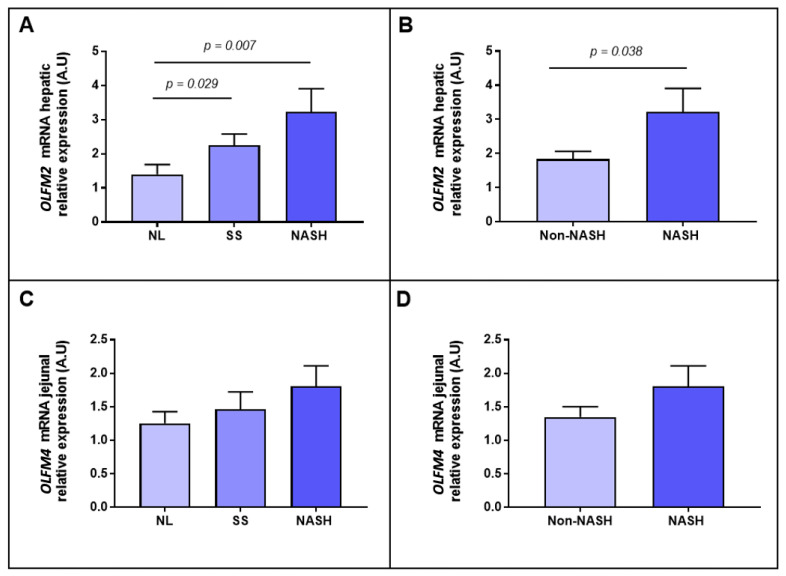
Differential relative mRNA abundance of *OLFM2* in hepatic tissue between women with MO (**A**) classified as NL, SS and NASH and (**B**) classified according to the presence or absence of NASH. Differential relative mRNA abundance of *OLFM4* in jejunal tissue between women with MO (**C**) classified as NL, SS and NASH and (**D**) classified according to the presence or absence of NASH. OLFM, olfactomedin; NL, normal liver; SS, simple steatosis; NASH, nonalcoholic steatohepatitis; A.U arbitrary units. Differences between groups were calculated using Mann–Whitney test and *p* < 0.05 was considered statistically significant.

**Figure 2 ijms-23-07442-f002:**
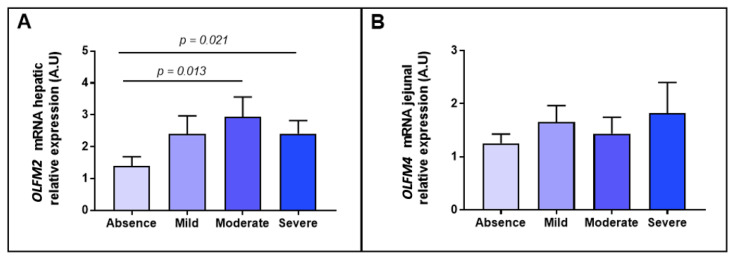
Differential relative mRNA abundance of (**A**) *OLFM2* in hepatic tissue and (**B**) *OLFM4* in jejunal samples between women with MO classified according to differences grades of steatosis into absence, mild, moderate and severe. OLFM, olfactomedin. A.U arbitrary units. Differences between groups were calculated using Mann–Whitney test and *p* < 0.05 was considered statistically significant.

**Figure 3 ijms-23-07442-f003:**
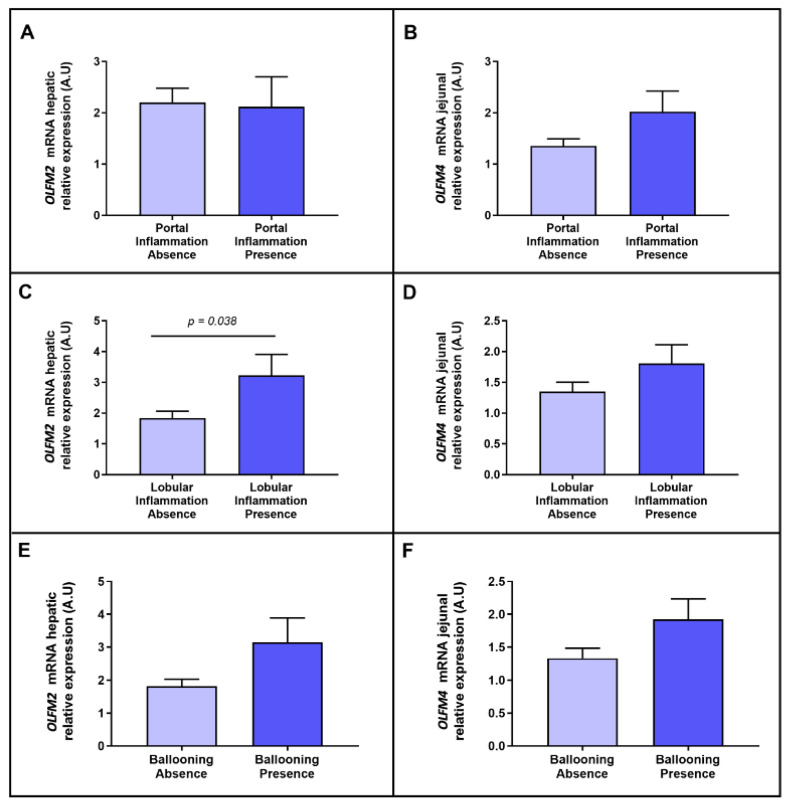
Differential relative mRNA abundance of hepatic *OLFM2* between, (**A**) absence or presence of portal inflammation, (**C**) lobular inflammation absence or presence and (**E**) ballooning absence or presence. Differential relative mRNA abundance of jejunal *OLFM4* between, (**B**) absence or presence of portal inflammation, (**D**) lobular inflammation absence or presence and (**F**) ballooning absence or presence. OLFM, olfactomedin. A.U arbitrary units. Differences between groups were calculated using Mann–Whitney test. *p* < 0.05 was considered statistically significant.

**Figure 4 ijms-23-07442-f004:**
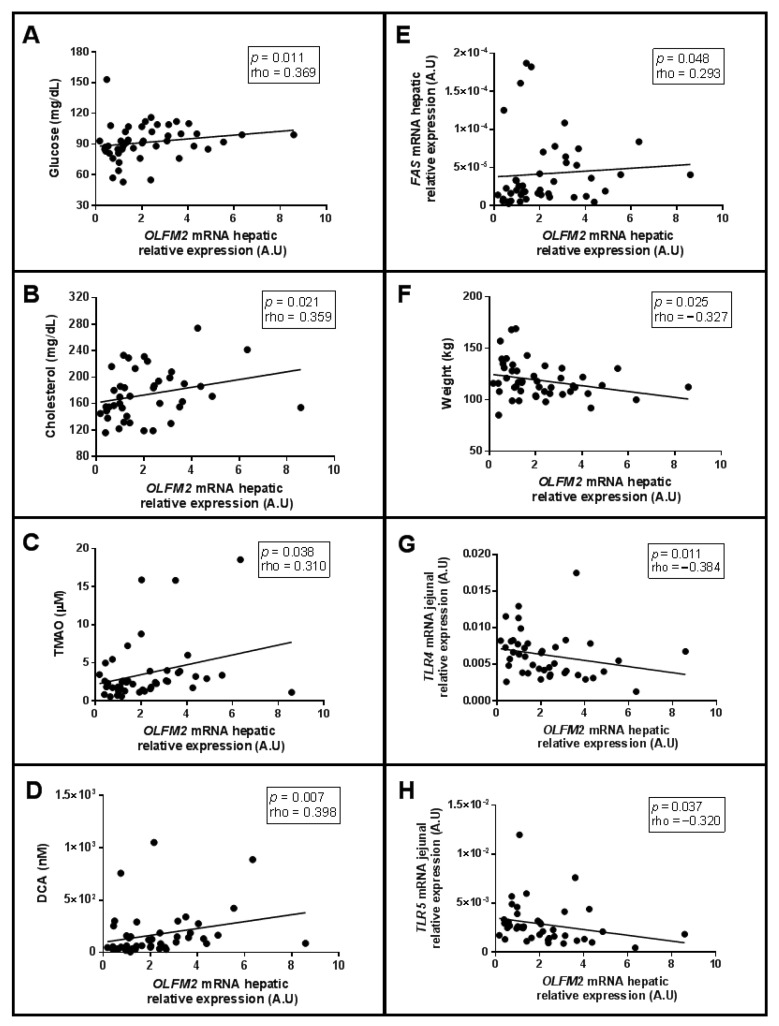
Significant correlations between *OLFM2* hepatic mRNA expression and (**A**) glucose, (**B**) cholesterol (**C**) TMAO, (**D**) DCA levels and (**E**) *FAS* in liver, (**F**) weight, (**G**) jejunal *TLR4* and (**H**) *TLR5* using Spearman’s method. OLFM, olfactomedin; TMAO, Trimethylamine N-oxide dihydrate; DCA, deoxycholic acid; TLR, Toll-like receptor FAS, fatty acid synthase; RE, relative expression; A.U, arbitrary units. Correlation coefficient (rho) was calculated using Spearman test. *p* < 0.05 was considered statistically significant.

**Figure 5 ijms-23-07442-f005:**
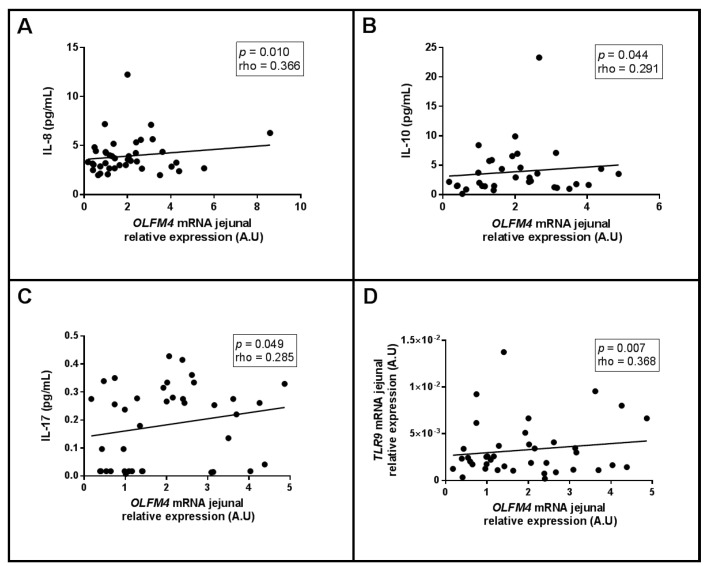
Significantly positive correlations between *OLFM4* jejunal mRNA expression (**A**) IL-8, (**B**) IL-10, (**C**) IL-17 levels and (**D**) *TLR9* in jejunum using Spearman’s method. OLFM, olfactomedin; TLR, Toll-like receptor; IL, interleukin; RE, relative expression; A.U, arbitrary units. Correlation coefficient (rho) was calculated using Spearman test. *p* < 0.05 was considered statistically significant.

**Table 1 ijms-23-07442-t001:** Anthropometric and biochemical variables of women in the studied cohort.

Variables	NL (*n* = 27)	SS (*n* = 26)	NASH (*n* = 16)
Weight (kg)	117.00 (107.00–131.00)	114.00 (108.98–128.60)	110.50 (104.33–120.75)
BMI (kg/m^2^)	43.50 (40.89–46.88)	44.35 (40.87–46.80)	44.19 (40.69–45.80)
SBP (mmHg)	120.00 (100.00–132.50)	117.50 (108.50–127.00)	115.00 (102.00–127.00)
DBP (mmHg)	63.00 (57.50–73.00)	62.00 (59.50–73.75)	64.00 (55.00–70.00)
HOMA1-IR	2.05 (1.03–3.45)	2.52 (1.38–3.68)	1.63 (1.26–4.23)
Glucose (mg/dL)	85.00 (76.00–93.00)	93.00 (87.25–107.00) *	91.50 (82.25–101.75)
Insulin (mUI/L)	9.57 (5.55–16.82)	10.17 (7.23–13.93)	7.19 (5.14–26.02)
HbA1c (%)	5.50 (5.30–5.70)	5.55 (5.30–5.95)	5.55 (5.15–6.13)
TG (mg/dL)	106.50 (94.00–136.00)	117.50 (82.25–172.50)	153.00 (116.50–256.50) *
Cholesterol (mg/dL)	170.00 (148.25–209.50)	171.15 (136.25–194.25)	183.90 (152.75–229.50)
HDL-C (mg/dL)	40.60 (32.05–48.50)	43.50 (33.75–47.00)	37.80 (33.50–48.50)
LDL-C (mg/dL)	107.90 (86.00–134.20)	104.10 (77.20–126.25)	94.00 (79.30–128.03)
AST (UI/L)	20.00 (15.50–36.50)	23.00 (17.00–35.00)	27.00 (17.25–43.50)
ALT (UI/L)	22.50 (16.00–37.50)	31.00 (22.00–32.25)	32.00 (16.25–41.00)
GGT (UI/L)	18.00 (15.25–26.25)	21.00 (16.00–32.25)	25.50 (18.00–28.75)
ALP (Ul/L)	58.50 (49.25–71.25)	74.00 (64.00–86.25) *	63.00 (55.00–74.50) ^$^

NL, normal liver; SS, simple steatosis; NASH, nonalcoholic steatohepatitis; BMI, body mass index; SBP, systolic blood pressure; DBP, diastolic blood pressure; HOMA1-IR, homeostatic model assessment method-insulin resistance; HbA1c, glycosylated haemoglobin; TG, triglycerides; HDL-C, high density lipoprotein cholesterol; LDL-C, low density lipoprotein cholesterol; AST, aspartate aminotransferase; ALT, alanine aminotransferase; GGT, gamma-glutamyltransferase; ALP, alkaline phosphatase. Data are expressed as the median (interquartile range). * Significant differences vs. NL group (*p* < 0.05). ^$^ Significant differences vs. SS group (*p* < 0.05).

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
