# Peer review of "New Insights of OLFM2 and OLFM4 in Gut-Liver Axis and Their Potential Involvement in Nonalcoholic Fatty Liver Disease"

_ijms, 2022, doi:10.3390/ijms23137442_

Round 1

Reviewer 1 Report

A well-written draft on the results of a study with an adequate design in parallel groups with morphological verification of the diagnosis of liver damage. Given the number of mediators which role in liver damage is discussed at various stages of pathogenesis, it is advisable to increase the number of participants in future studies.

Reviewer 2 Report

Title: New insights of OLFM2 and OLFM4 proteins in gut-liver axis and their potential involvement in nonalcoholic fatty liver disease

Authors:

Laia Bertran, Rosa Jorba-Martin, Andrea Barrientos-Riosalido, Marta Portillo-Carrasquer, Carmen Aguilar, David Riesco, Salomé Martínez, Margarita Vives, Fàtima Sabench , Daniel Del Castillo, Cristóbal Richart and Teresa Auguet

General comment:

Nonalcoholic fatty liver disease (NAFLD) is increasingly common worldwide; in some populations, it is the most common form of chronic liver disease. The clinical course of NAFLD is hardly predictable; therefore, it is challenging to assess which patients develop nonalcoholic steatohepatitis (NASH) and, finally, cirrhosis and liver failure. Therefore there is a need for novel markers that would facilitate the diagnosis of NAFLD and enable the identification of individuals at high risk of NASH. In their work, Laia Bertran et al. looked for the possible role of olfactomedins (OLFMs) in the liver and gut expression in the pathogenesis of NAFLD. Despite its descriptive character, the manuscript is well designed and written; therefore, I have only some minor remarks that should be addressed before it is accepted for publication.

Minor revisions:

1)      Abstract:

"Our results showed that OLFM2 hepatic mRNA was higher in NASH, in advanced degrees of steatosis and in the presence of lobular inflammation presence.” – please remove the repetition.

2)      Introduction:

“Nonalcoholic fatty liver disease (NAFLD) is characterized by ectopic fat accumulation in hepatic tissue in the absence of secondary causes of steatosis, such as alcohol intake." – please add "and/or viral infection."

“OLFM2 seems to have a role in the regulation of energy, and OLFM4 may participate 93 in gastrointestinal inflammation in relation to dysbiosis.” – please be more specific – the word “energy” may refer to anything.

3)      Results

Figures 4 &5  – please correct the figures’ legends – the results represent correlations, so the sentence “Differences between groups were calculated using Mann-Whitney test.” seems inappropriate.

4)      Discussion

Please provide some explanation for finding no association between gut OLFM4 expression and the severity of liver steatosis and speculate on potential confounders that could have influenced the results.

Please suggest what kind of further studies should be performed to establish the role of OLFMs in the pathogenesis of NAFLD and their relation-325 ship with the gut-liver axis and inflammation.

5)      Materials and Methods

“The exclusion criteria were as follows: (1) an intake of ethanol higher than 10 g/day or other toxins;  (2) patients who had infectious disease or neoplastic disease or an acute or chronic hepatic disease different from NAFLD; (3) menopausal women or women using contraceptives; (4) patients treated with fibrates.” – to make this section more informative please justify the exclusion criteria 3 & 4.

Please add information on what statistical test was used to perform correlation analyses.

Whole manuscript:

1)      Please explain the abbreviations as they only appear in the text (e.g., NL, SS) to facilitate the perception of the manuscript.

2)      Please check the manuscript profoundly to verify when OLFM refers to the gene’s name (and should be written in italics) and to the protein name.
